# Molecular structure of promoter-bound yeast TFIID

Olga Kolesnikova [1,2,3,4], Adam Ben-Shem[1,2,3,4], Jie Luo[5], Jeff Ranish [5], Patrick Schultz [1,2,3,4] & Gabor Papai [1,2,3,4]

Transcription preinitiation complex assembly on the promoters of protein encoding genes is nucleated in vivo by TFIID composed of the TATA-box Binding Protein (TBP) and 13 TBP-associate factors (Tafs) providing regulatory and chromatin binding functions. Here we present the cryo-electron microscopy structure of promoter-bound yeast TFIID at a resolution better than 5 Å, except for a flexible domain. We position the crystal structures of several subunits and, in combination with cross-linking studies, describe the quaternary organization of TFIID. The compact tri lobed architecture is stabilized by a topologically closed Taf5-Taf6 tetramer. We confirm the unique subunit stoichiometry prevailing in TFIID and uncover a hexameric arrangement of Tafs containing a histone fold domain in the Twin lobe.

---

[1] Department of Integrated Structural Biology, Equipe labellisée Ligue Contre le Cancer, Institut de Génétique et de Biologie Moléculaire et Cellulaire, Illkirch 67404, France. [2] Centre National de la Recherche Scientifique, UMR7104, 67404 Illkirch, France. [3] Institut National de la Santé et de la Recherche Médicale, U1258, 67404 Illkirch, France. [4] Université de Strasbourg, Illkirch 67404, France. [5] Institute for Systems Biology, Seattle, WA 98109, USA. These authors contributed equally: Olga Kolesnikova, Adam Ben-Shem. Correspondence and requests for materials should be addressed to P.S. (email: patrick.schultz@igbmc.fr) or to G.P. (email: gabor.papai@igbmc.fr)

Transcription of eukaryotic protein encoding genes is initiated by the recruitment of general transcription factors together with RNA polymerase II to gene promoters to form the transcription preinitiation complex (PIC)[1,2]. In vitro PIC assembly begins with the recognition of a specific TATAA promoter sequence by the TATA-box Binding Protein (TBP) which is part of the TFIID complex along with 13 conserved TBP-associated factors (Tafs)[3]. TBP contributes in positioning the PIC at a fixed distance from the transcription start site at TATA-containing promoters while a less precise definition of the transcription start site is observed at TATA-less promoters. In higher eukaryotes, additional conserved DNA motifs contribute to promoter recognition by TFIID[4–6] and some Tafs assist TBP in promoter recognition[7,8]. Within TFIID, TBP is locked in an inactive state where its DNA binding activity is repressed through an interaction of the N-terminal domain of Taf1 with the concave DNA-binding surface of TBP. This auto-inhibition can be relieved by TFIIA that considerably stabilizes TFIID promoter interaction[9–12]. For regulated in vivo gene transcription, the TFIID complex binds to gene promoters through combined interactions with promoter DNA, transcriptional activators and specific epigenetic histone modifications[13].

A striking feature of TFIID's structural organization is that nine Tafs contain a stretch of amino acids with sequence homology to histones, including the histone fold (HF) domain involved in histone dimerization[14]. These homologies were confirmed by X-ray diffraction studies which revealed that drosophila Taf9 and Taf6 form a heterotetramer and interact through a characteristic histone fold[15] and that human Taf11 and Taf13[16], as well as Taf4 and Taf12[17] also contain a HF used to form heterodimers. Sequence alignment, specific heterodimerization of bacterially coexpressed Tafs and two-hybrid assays, showed that also Taf3-10 and Taf8-10 can form specific heterodimers[18,19].

A very unique subunit stoichiometry prevails in TFIID since a subset of six subunits are present in two copies (Taf5, 6, 9, 4, 12, 10), while the remaining seven Tafs are present as single copies. A TFIID core containing two copies of Tafs 5, 6, 9, 4 and 12, was produced in insect cells and its cryo-EM structure showed a clear two-fold symmetry which was broken by the addition of the Taf8-10 heterodimer[20].

Yeast TFIID contains an additional subunit, Taf14[12], which is also a constituent of six other transcription related complexes thus making its role as a bona-fide TFIID subunit difficult to evaluate. Although Taf14 is not conserved as a TFIID subunit in metazoans, its function may be retained through the chromatin binding YEATS domains that may substitute for some of the missing metazoan chromatin interaction domains such as the human TAF1 double bromodomain and the TAF3 plant homeobox domain (PHD)[21,22]. Genetic interactions of Taf14 with Taf2 were described and biochemically mapped to the C-terminus of Taf2[23].

Despite extensive efforts, TFIID is still poorly understood at the structural level and no atomic model of the full complex is currently available. Inherent flexibility, poor complex stability, and sub-stoichiometric subunit composition prevented reaching high resolution structural information. Cryogenic electron microsopy (cryo-EM) and footprinting studies indicated that human TFIID can undergo massive structural rearrangements[24]. TFIID was shown to co-exist in two distinct structural states while the presence of both TFIIA and promoter DNA stabilizes a rearranged state of TFIID that enables promoter recognition and binding. A recent breakthrough was achieved by stabilizing human TFIID through its binding to TFIIA and a chimeric super core promoter (SCP) designed by combining several core promoter binding motifs found in metazoans[25]. This structure enabled the fitting of the atomic coordinates of TAF2, TAF1 and

two copies of the TAF6 HEAT repeats. It also showed the interaction of TAF2 with downstream DNA promoter elements, and revealed DNA-bound TBP in a remote position where it interacts only with TFIIA.

Here we describe the cryo-EM structure of the yeast *Komagataella phaffii* (formerly known as *Pichia pastoris*) TFIID complex at a resolution of 4.5 to 10.7 Å. We position the crystal structures of several subunits and, in combination with cross-linking studies, describe the quaternary organization of TFIID. The compact tri lobed *Kp*TFIID architecture is stabilized by a topologically closed (Taf5-Taf6)$_2$ tetramer. We confirm the unique subunit stoichiometry prevailing in TFIID and uncover a hexameric arrangement of Tafs containing a HF domain in the Twin lobe. Interaction with promoter DNA highlights two non-selective binding sites consistent with a DNA scanning mode.

## Results

**Structure of TFIID.** Currently available structural information on the complete yeast TFIID complex is derived from low-resolution electron microscopy studies precluding docking of existing atomic models[26,27]. We developed a TFIID purification protocol from the yeast *K. phaffii* for structure determination by high-resolution cryo-EM. SDS-PAGE experiments (Fig. 1a) and mass spectrometry analysis (Supplementary Data 1) confirm that the TFIID subunit composition of the phylogenetically closely related *S. cerevisiae* and *K. phaffii* are identical. To gain a better understanding of the subunit organization within TFIID, a crosslinking mass spectrometry (CXMS) analysis was performed using the homo-bifunctional, amine-reactive crosslinking reagent bis(sulfosuccinimidyl)suberate (BS3). A total of 391 unique intra-protein crosslinks (intralinks) and 488 unique inter-protein crosslinks (interlinks) were identified (Supplementary Data 2). The dense BS3 crosslinking map underlined many known protein-protein and domain-domain interactions within the TFIID complex (Fig. 1b, c, and Supplementary Fig. 1). For example, Taf7 is extensively crosslinked to Taf1, in agreement with previous studies showing that these two subunits directly interact[28]. Genetic and biochemical evidence indicated that Taf5, Taf4, Taf12, Taf6 and Taf9 are present as two copies in TFIID[12] and a recombinant human core TFIID containing these 5 Tafs shows two-fold symmetry[20]. Taf5 crosslinks extensively to all the core subunits (Fig. 1c and Supplementary Fig. 1), suggesting that this subunit serves as an organization center for core subunits assembly. Nine Tafs contain a protein fold homologous to nucleosomal HF domain which were shown to form 5 specific heterodimers in vitro[16,29,30]. With the exception of the Taf3-10 HF domain pair, all other heterodimer partners were found crosslinked (Taf11-13, Taf4-12, Taf6-9 and Taf8-10). TBP crosslinks with several documented interacting partners in the TFIID complex. The N-terminal TAND domain of Taf1 was reported to interact with TBP to inhibit its binding to the TATA-box[11]. The Taf11-Taf13 heterodimer was recently shown to interact with TBP and to compete with the Taf1 TAND domain for TBP binding[31]. The same TBP residues were observed to crosslink with both Taf11-Taf13 and Taf1 N-terminus, indicating that within holo-TFIID, TBP may exist in distinct states and interact with multiple competing partners.

Single particle analysis of mildly crosslinked, frozen hydrated *Kp*TFIID molecules resulted in a 3-D reconstruction with an overall resolution of 12.1 Å (Fig. 2a). The compact TFIID structure is composed of three lobes interconnected by protein bridges thus forming a closed ring-like system. This arrangement fits with early observations of negative stained particles[26] but differs from cryo-EM studies featuring extended or horseshoe shaped models[27]. Small improvements in various steps of

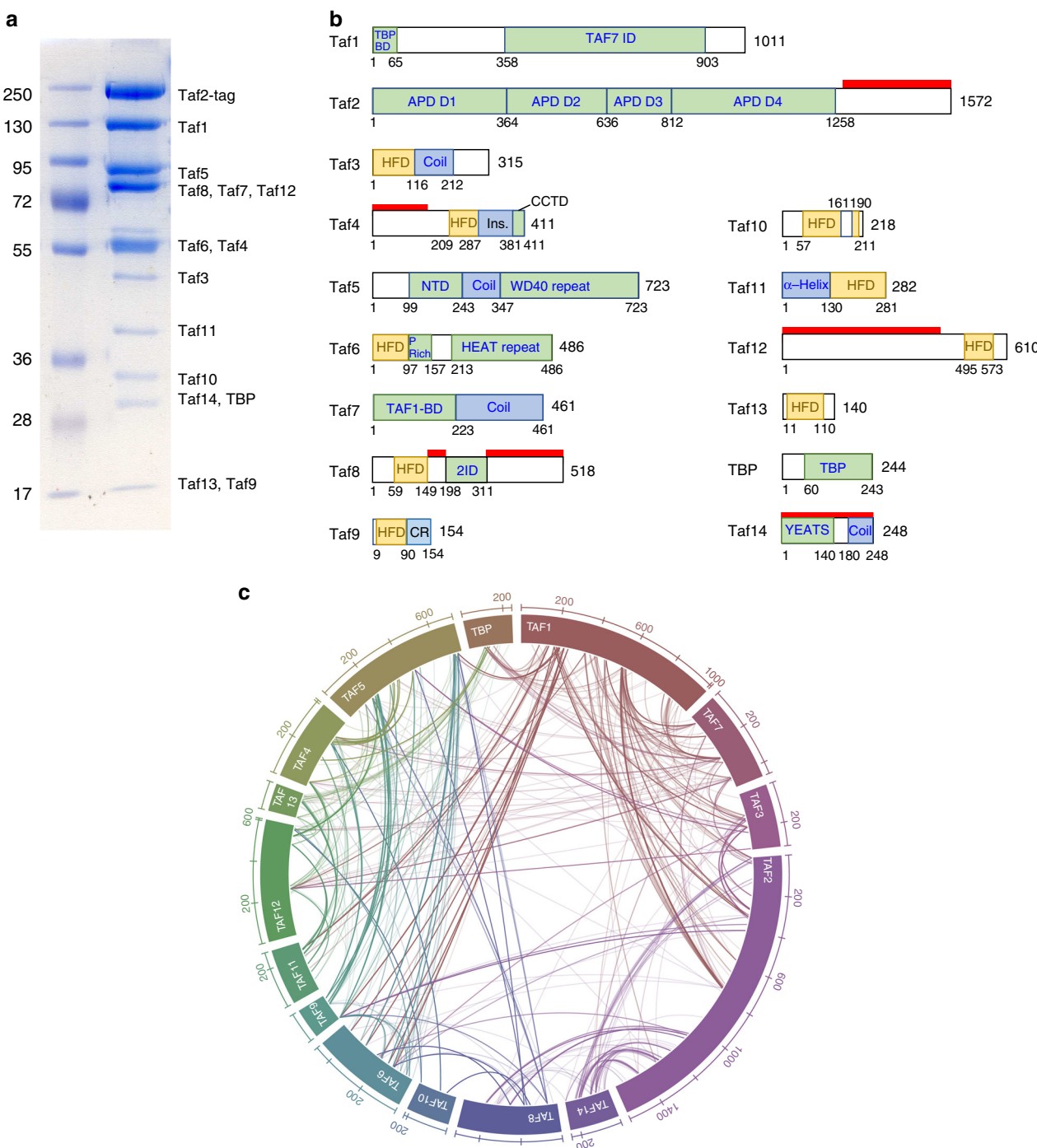

**Fig. 1** Purification and interaction map of yeast TFIID. **a** Colloidal coomassie blue stained SDS-PAGE analysis of TFIID complex purified from the SBP-tagged Taf2 strain. **b** Schematic representation of the TFIID subunits showing the conserved structural domains (colored boxes) and the yeast specific domains (red bars). TBP-BD: TBP binding domain (TAND), Taf7 ID: Taf7 interacting domain, APD: aminopeptidase, HFD: Histone fold domain, INS: Insertion, CCTD: conserved C-terminal domain, NTD: N-terminal domain, WD40: structural motif of approximately 40 amino acids, often terminating in a tryptophan-aspartic acid (W-D) dipeptide, HEAT: structural motif composed of two alpha helices linked by a short loop, P-rich: proline rich domain, TAF1-BD: Taf1 binding domain, 2ID: Taf2 interacting domain, CR conserved HFD flanking region, YEATS: Yaf9, ENL, AF9, Taf14, Sas5 domain. **c** Subunit-subunit cross-linking map. Line transparency corresponds to the number of cross-links identified between the two subunits

specimen preparation may have stabilized the linkers between the three lobes. To further stabilize TFIID and reach higher resolution, the complex was incubated with TFIIA and the 105 bp TATA-box containing *K. phaffii* glyceraldehyde-3-phosphate dehydrogenase promoter (pGAP). Electrophoretic mobility shift experiments show that TFIID at a concentration of 0.4 μM interacts with DNA (0.2 μM) and shifts about half of the DNA in the absence of TFIIA. The addition of TFIIA (0.8 μM) modifies slightly the electrophoretic mobility of the complex (Supplementary Fig. 2).

**Promoter-bound TFIID.** Single particle cryo-EM analysis of the promoter-bound complex combined with local map refinement showed a significant resolution improvement for the two lobes interacting with DNA while the third one remained poorly

resolved (Fig. 2b and Supplementary Fig. 3). The overall 3 lobed structure of the TFIID complex was not drastically reorganized upon promoter DNA binding, except for minor conformational changes within the poorly resolved Taf1 lobe. Based on the cryo-EM structures and the crosslinking data, we named the yeast TFIID lobes: Taf1 lobe, Twin lobe and Taf2 lobe, previously named lobe A, B and C, respectively.

The Taf2 lobe was resolved to 4.5 Å resolution thus revealing the secondary structure of the N-terminal aminopeptidase homology domain of Taf2 (Fig. 3a) consistent with the recently published structure of human TFIID[25]. An atomic model of Taf2 residues 36 to 1194, including the 4-subdomain arrangement of the aminopeptidase domain (D1-D4) could be fitted into the cryo-EM map. Recombinant human TAF2 was shown to form a stable complex with the HF domain-containing TAF8-TAF10 heterodimer[30]. The non-HF domain C-terminal part of hTAF8 is crucial to interact with hTAF2. Secondary structure predictions foresee an evolutionarily conserved α-helix placed after the HF domain and a short proline rich region (residues 234–259 in *Kp*Taf8) (Supplementary Fig. 4). In human TFIID, this helix was proposed to insert between the TAF6 HEAT repeat and TAF2. A helix density is located at the same place in *Kp*TFIID, and this helix crosslinks to both the Taf6 HEAT repeat and to Taf2 D4 (Supplementary Fig. 4). Accordingly, its attribution to the conserved Taf8 helix is plausible. The Taf8–10 HF domain dimer could however not be located in the Taf2 lobe density. In most yeast species Taf8 genes have an additional C-terminal extension in which two long α-helices are predicted (Supplementary Fig. 4). Two such long helices are visible in the cryo-EM map at the external surface of Taf2 and their attribution to the C-terminal part of Taf8 is strengthened by their absence in the human TFIID

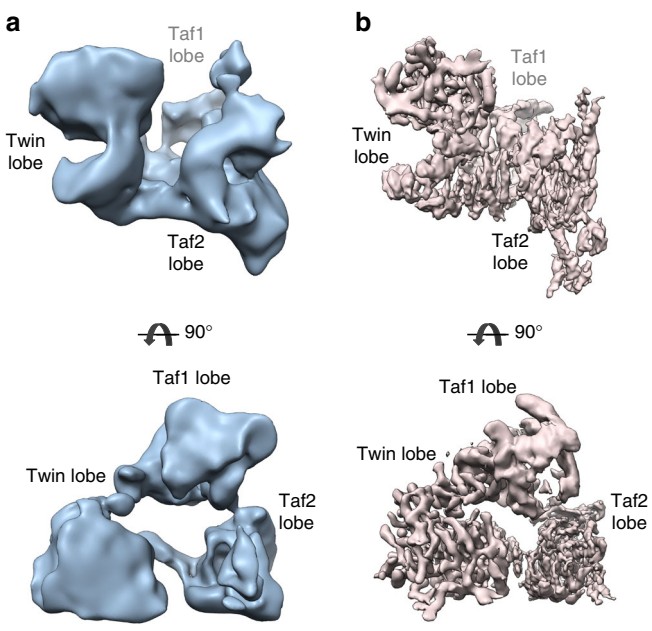

**Fig. 2** Structural organization of yeast TFIID **a** Cryo-EM model of the yeast *Komagataella phaffii* TFIID at a resolution of 12.1 Å. **b** Cryo-EM model of the TFIID-TFIIA-pGAP complex

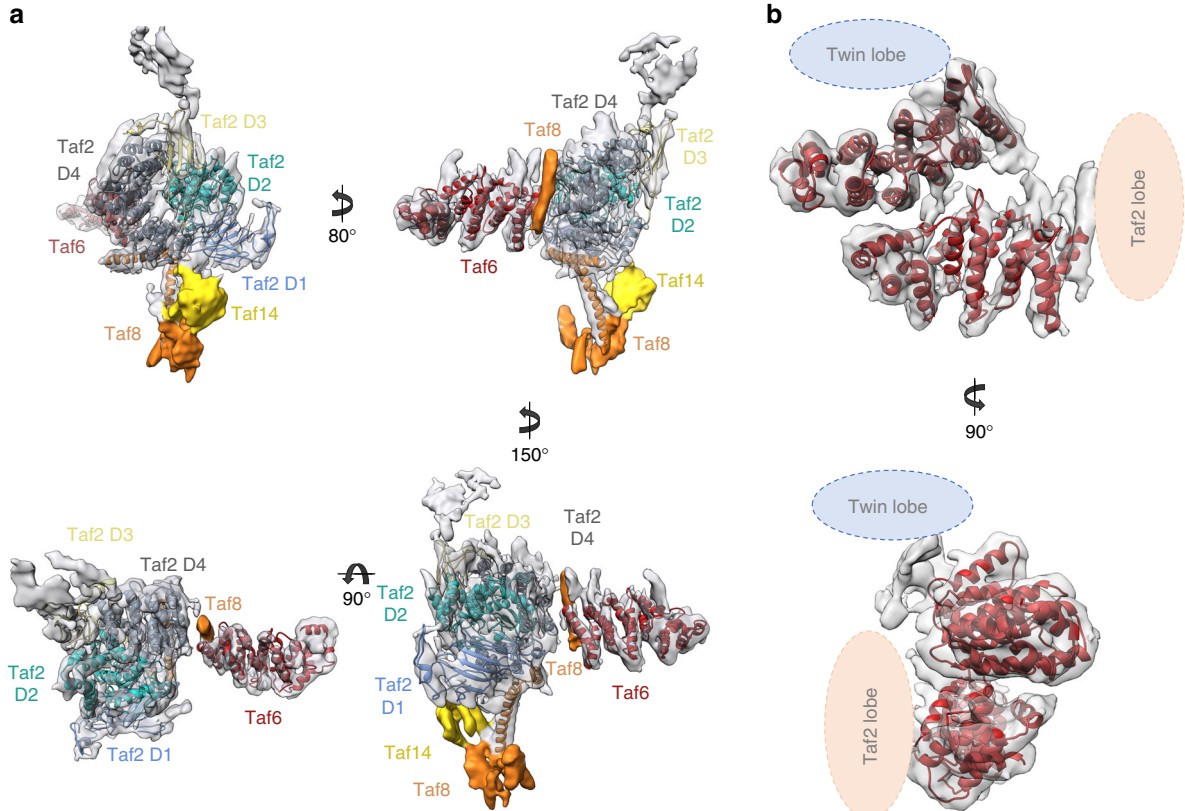

**Fig. 3** Subunit arrangement within the Taf2 lobe **a** Atomic model docking of the amino-peptidase-like domains of Taf2 (from D1 to D4), the predicted Taf8 helices, and the Taf14 YEATS domain within the Taf2 lobe reconstructed at 4.5 Å resolution. **b** Linker between the Taf2 lobe and the Twin lobe consisting of two Taf6 HEAT repeats

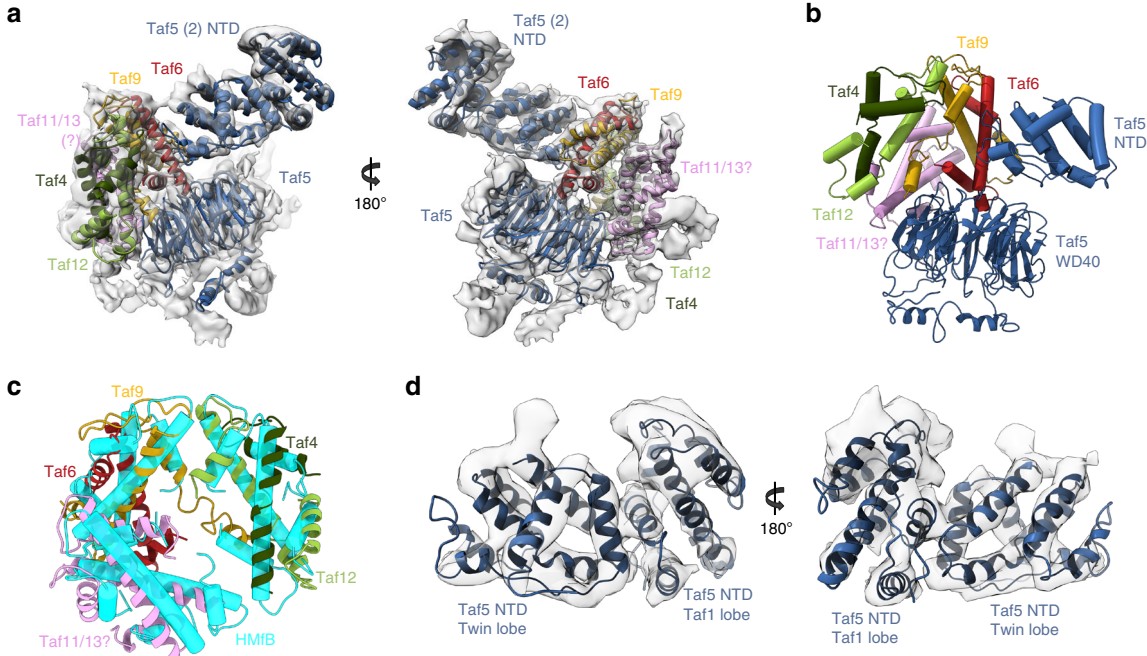

**Fig. 4** Subunit arrangement of the Twin lobe **a** Atomic model docking of the Taf5 WD40 repeat, Taf5 NTD, Taf6-9, Taf11-13 and Taf4-12 HF domain heterodimers within the Twin lobe. **b** Central role of the Taf5-WD40 repeat in organizing the HF domain heterodimers. **c** Structural homology between the Taf6-9-4-12-11-13 HF domain hexamer (ribbons) and the archaeal histone hexamer (tubes). **d** Linker between the Twin lobe and the Taf1 lobe consisting of two Taf5 NTDs

structure[25]. One Taf8 helix contacts two Taf2 helices formed by residues 1071–1081 and 1106–1118, respectively, supported by multiple crosslinks between Taf2 1068–1104 to Taf8 206–391. The second Taf8 helix contributes to a protein stalk protruding out of the Taf2 lobe and terminated by a protein bulge representing most probably the C-terminal end of Taf8. A compact protein density is observed at the base of this yeast-specific Taf8 stalk. The size and shape of this density is consistent with the N-terminal YEATS domain of Taf14, a subunit only found in yeast. The YEATS domain is densely crosslinked to the C-terminal end of the aminopeptidase region of Taf2, while the C-terminal coil domain of Taf14 crosslinks to the N-terminal region (1–118) of Taf2. Both regions of Taf2 are located close to the stalk base. These results agree with a previous study showing that a mutant *Sc*TFIID lacking the yeast-specific Taf2 C-terminal residues 1260–1407 is unable to interact with Taf14[23]. Furthermore, thermosensitive mutations in Taf2, whose phenotype is suppressed by overexpressing Taf14, were also positioned in the vicinity of this density. Altogether these observations indicate that the stalk is formed by Taf14 and the C-termini of Taf8 and Taf2, thus defining a yeast-specific module absent from human TFIID. This module may provide yeast TFIID with a chromatin binding module that was suggested to be replaced in higher eukaryotes by the Taf1 double bromodomain and the Taf3 plant homeobox domain[23].

The linker between the Taf2 lobe and the Twin lobe is built from two copies of Taf6 HEAT repeats (residues 213–486), thus confirming that Taf6 is present twice in holo-TFIID (Fig. 3b). The HEAT repeats are placed at an angle of 40° with a twist of 80° between them and their C-termini are exposed to the periphery of TFIID. Despite their interaction, no symmetry was found between the repeat arrangements. The two-fold symmetry of a recombinant human core TFIID containing two copies of the TAF5, TAF4, TAF12, TAF6 and TAF9 subunits was shown to be compromised by the binding of a single copy of the TAF8-TAF10

heterodimer[20]. The above described putative Taf8 helix, associated with the HEAT repeat closest to the Taf2 lobe, may play a role in introducing asymmetry upon binding to core TFIID.

Connected to the second Taf6 HEAT repeat, the Twin lobe was resolved to a resolution of 4.8 Å thus allowing the identification of most alpha helices (Fig. 4a). A crystal structure of the human Taf5 WD40 propeller and NTD together with the Taf6-Taf9 HF domain heterodimer (courtesy of Dr Imre Berger, unpublished data) fits remarkably well with our cryo-EM structure and helped us to position the WD40 repeat domain of Taf5. Unexpectedly we could identify two additional HF domain heterodimers in the remaining density. The WD40 domain forms the core of the Twin lobe and coordinates the arrangement of the three HF domain heterodimers (Fig. 4b). Two of these were attributed to the Taf6-9 and Taf4-12 pairs since these subunits form a stable subcomplex with Taf5[20] and are extensively crosslinked to Taf5. Taf11 and Taf13 are the best candidate to form the third Taf5-bound HF domain pair, although we cannot exclude the possibility that the third HF domain heterodimer is formed by Taf8 and Taf10. Taf11 and Taf13 strongly crosslink to all partners of this lobe and fail to crosslink with specific subunits of other lobes such as Taf2 or Taf1, except for the highly flexible N-terminal part of Taf1. In the X-ray crystal structure of the human TAF4-TAF12 histone-like heterodimer the α3 helix of the predicted Taf4 HF domain was missing. It was suggested that the α3 helix is separated from the α2 helix by an extended loop and that the characteristic fold would reconstitute with the full Taf4 protein[17]. Our results show that the α3 helix of Taf4 is absent in the TFIID complex and thus confirm the unconventional nature of the Taf4-Taf12 heterodimer (Supplementary Fig. 5). Functional and biochemical analysis of yTaf4 provided strong evidence that the conserved C-terminal domain (CCTD) of Taf4 and the linker adjacent to the histone-fold domain contribute to Taf4-Taf12 heterodimer stability and contains a conserved functional domain essential

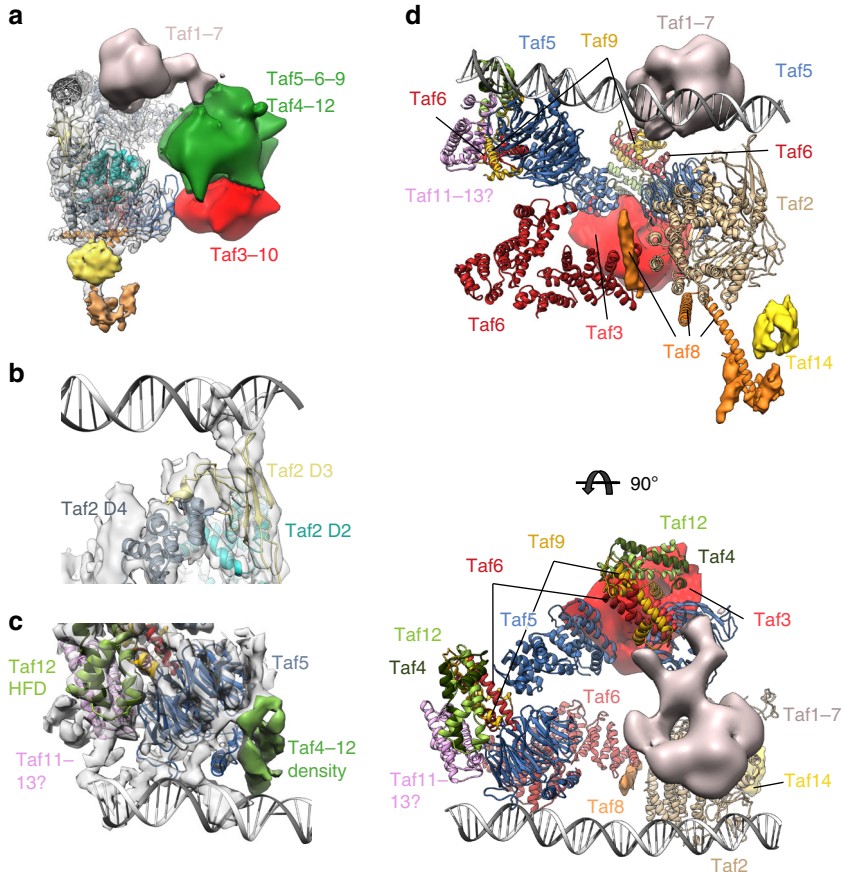

**Fig. 5** Predicted organization of the Taf1 lobe and DNA interactions **a** Organization of the Taf1 lobe as derived from Volta Phase Plate images of frozen hydrated TFIID-TFIIA-pGAP complexes. The position of the second Taf5-6-9-4-12 module could be determined (green domain). The putative Taf1 flexible domain (grey) is facing the Taf2-bound DNA. **b** TFIID-DNA interactions within the Taf2 lobe. **c** TFIID-DNA interactions within the Twin lobe. **d** Proposed arrangement of the fitted atomic models and subunits in yeast TFIID

for yeast growth[29]. The non-attributed densities in the Twin lobe are likely to arise from Taf12 and Taf4 N-terminal extensions.

The protein density linking the Twin lobe with the flexible Taf1 lobe is formed by a dimer of Taf5 NTD (residues 99–243)[32] thus confirming that TFIID contains two copies of Taf5[12,20,26] (Fig. 4d). The two NTD domains are not symmetry related in holo-TFIID but reproduce the arrangement of the asymmetric unit found in crystals of this domain[32].

The Taf1 lobe map shows poor resolution in part due to conformational heterogeneity, which could not be sorted out by local classification or refinement, possibly due to the weak contrast of cryo-EM images. To overcome this limitation, images of the TFIID-TFIIA-pGAP complex were recorded with a Volta Phase Plate (VPP) to produce highly contrasted images[33] (Supplementary Fig. 6). After local alignment, the Taf1 lobe appears as a large globular domain (green and red in Fig. 5a) from which an extended protein domain terminated by a bulge protrudes out (grey in Fig. 5a). The globular domain contacts the N-terminal Taf2-D1 domain and contains a ring like-structure (Supplementary Fig. 7) corresponding most probably to the second Taf5 WD40 repeat as expected by the stoichiometry of Taf5. Since Taf5 and Taf6 are found in two copies in holo-TFIID, we propose that a second Taf5-9-6-4-12 module, similar to the one forming the Twin lobe, is present in the Taf1 lobe (Green in Fig. 5a). We further suggest that a Taf3-Taf10 HF domain heterodimer completes in the Taf1 lobe the hexameric HF domain-structure found around the Taf5 WD40 repeat in the Twin lobe. Such a position for the Taf3-Taf10 HF pair is strongly

supported by the CXMS data (red in Fig. 5a–d, Supplementary Fig 7b). The VPP image analysis disclosed a large protein density extending from the Taf1 lobe and pointing towards Taf2 (grey in Fig. 5a). This flexible arm is also detected without VPP but only when the density threshold is lowered. This domain is likely to correspond to Taf1 and/or Taf7 since the surface of Taf2 facing this domain cross-links preferentially with these two subunits (Supplementary Fig. 7). The crystal structure of a human complex comprising the highly conserved central and amino-terminal fragments of Taf1 and Taf7[34], respectively, was placed in a similar position in the human TFIID complex[25].

A continuous thread of density suitable to accommodate a double stranded DNA molecule is located between the Taf2- and the Twin lobes only in the promoter-bound complex (Supplementary Fig. 8a). The signal is weak except at sites were DNA interacts with TFIID, suggesting a low DNA occupancy or an important DNA flexibility. The promoter DNA contacts Taf2 through two arginine and lysine rich loops of the D3 domain of the aminopeptidase domain (residues 733–742 and 646–652) (Fig. 5b). The pGAP DNA is clamped between Taf2 and the Taf1/Taf7 arm but the Taf1/Taf7 arm does not interact directly with the pGAP promoter. In the Twin lobe the DNA path is distant from the HF domain-containing Taf hexamer consistent with the observation that the side chains that mediate contacts between nucleosomal histones and DNA have not been conserved[17]. The DNA interacts with the N-terminal regions of Taf4 and/or of Taf12 consistent with the reported non-sequence-specific in vitro DNA binding activities of these subunits[35,36] (Fig. 5c).

## Discussion

The present study contributes to our understanding of the molecular architecture of the general transcription factor TFIID by resolving the secondary structure of several Taf subunits. In solution, the yeast complex adopts a compact 3 lobed structure connected by three well resolved linkers. This overall organization is consistent with previous models obtained in negative stain which however showed a gap between the Taf1 and the Twin lobes[26,37]. This gap was even more pronounced in our previous cryo-EM map which showed an open, horseshoe shaped arrangement[27,38], as well as in the human TFIID structure which adopts an extended conformation[39]. This compact organization is maintained by the (Taf6-Taf5)$_2$ tetramer which forms a topologically closed protein ring running through the three lobes (Fig. 5d and Supplementary Fig. 9). The more extended conformations that were observed may arise from the disruption of the Taf5-NTD dimerization interaction. This topology provides robustness to the TFIID structure and may explain that the removal of the Taf6-HEAT repeats which connect the Taf2 lobe to the Twin lobe only moderately affects complex stability[40]. Such a compact architecture of the yeast complex leaves however little space for a major subunit rearrangement between the TFIID lobes, as was described for the human TFIID[24]. Human TFIID was found to adopt two rearranged states in which one lobe (human lobe A) can be associated with either lobe B or lobe C. Furthermore, the presence of both TFIIA and promoter DNA was shown to stabilize one particular rearranged state that enables promoter recognition and binding. In the yeast system, such a rearrangement is not observed and the compact TFIID structure is not affected upon DNA binding.

In vivo self-association studies, quantitative gel electrophoresis profiles and immunoelectron microscopy experiments showed that 7 TFIID subunits are present in more than one copy within the purified TFIID complex. Our structural data confirm directly that Taf5 and Taf6 heterodimer are present in two copies in the native complex. The poor resolution of the flexible Taf1 lobe prevents direct recognition of molecular folds, but we can infer that the HF domain-containing Taf6-9 and Taf4-12 heterodimers are present in two copies. We previously described the dimeric arrangement of a recombinant human core TFIID complex containing 5 human TAFs (TAF5, TAF6-9 and TAF4-12) and presenting a two-fold symmetry[20]. Strikingly, the molecular interactions between core subunits are completely reorganized in the TFIID complex and the two-fold symmetry has been lost. The Taf6 HEAT repeats and the Taf5 N-terminal domains interact in the mature TFIID and form well defined bridges between the lobes, while they are separated in core TFIID. The HF domains of Taf6/9 and Taf4/12 interact with the Taf5 WD40 repeats in full TFIID while this was not the case in core TFIID. The heterotetrameric arrangement of the Taf5-Taf6 core-TFIID subunits adopt an extended circular structure within TFIID that would probably not be stable without interactions with other Tafs, thus supporting the hypothesis that a massive rearrangement takes place upon TFIID maturation. Such rearrangements have been observed upon addition of the TAF8/TAF10 heterodimer to the core TFIID which started to lose internal symmetry[20].

Sequence analysis, in vitro interaction data and structural studies showed that nine Tafs contain a histone-like fold allowing the formation of 5 distinct heterodimers and a total of 7 HF pairs when considering the subunit copy number. The analysis of the TFIID density map detected directly 3 heterodimers in the Twin lobe and predicted 3 pairs in the Taf1 lobe. The predicted Taf10-Taf3 pair could not be confirmed by the current structural analysis due to limited resolution in the Taf1 lobe. Biochemical and structural data suggested a similarity of Taf4 and Taf12 to histones H2A and H2B, and of Taf6-Taf9 to histones H3 and H4, respectively, leading to the proposal that a histone octamer-like structure may exist in TFIID. The cryo-EM structure of yeast TFIID rules out the possibility that Taf4, Taf6, Taf9 and Taf12 form a histone octamer-like arrangement but revealed instead a hexameric arrangement. A similar arrangement was recently described for archaeal histone homodimers[41] where the small basic HMfB proteins, which share a common ancestor with the eukaryotic core histones and are able to interact with DNA by forming a trimeric arrangement of (HMfB)$_2$ homodimers. The structure of three (HMfB)$_2$ dimers and of the HF domain-containing Tafs is highly similar to the nucleosome hexasome, obtained by removing one H2A–H2B heterodimer from the nucleosome structure (Fig. 4c). Such an hexameric HF domain-Taf architecture was not reported to assemble in vitro thus emphasizing the key role played by the WD40 repeat of Taf5 in holding together the heterodimers.

In metazoan TFIID, Taf1 and Taf2 bind to the conserved initiator (INR) core promoter motif[42,43], and Taf6 together with Taf9 interact with the downstream promoter element[44]. The binding of specific Tafs to these conserved promoter DNA sequence motifs produces sharp DNase I protections and contributes to a strong and specific interaction of metazoan TFIID with promoters. Neither the INR nor the downstream promoter element have been unambiguously identified in the yeast system[45]. ScTFIID histone fold pairs Taf4-Taf12 and Taf6-Taf9 also display in vitro DNA binding activities, but this interaction has not been shown to be sequence-specific[35]. Structural analyses with ScTFIID-TFIIA-activator in complex with promoter-DNA position DNA in contact with the C terminus of Taf2[27]. However, these interactions of promoter DNA with ScTFIID do not produce sequence specific footprints and the TBP footprint on the TATA-box is predominantly observed[12].

The DNA path is similar in the human TFIID-TFIIA-SCP complex indicating that the basic DNA interaction modalities are conserved throughout evolution[25] (Supplementary Fig. 8 and 10). This binding mode is characterized by two distinct DNA-Taf interaction sites located, respectively, in the Taf2 lobe and in the Twin lobe. The Taf6-HEAT linker keeps the Taf2 and the Twin lobe DNA interaction sites at a constant distance (Fig. 5d). This bi-partite DNA binding architecture suggests that TFIID could scan the promoter DNA and facilitate the binding of TBP leading to PIC formation. Yeast TFIID exposes a stretch of 35–40 base pairs of DNA separating the two contact sites. This distance is larger than the yeast 18 bp mean nucleosome linker length[46] suggesting that in this DNA binding mode, the fixed distance could help to select nucleosome free promoter regions and recruit TFIID. The DNA-binding Twin lobe stands out as a key regulatory platform for TFIID function. Taf11 and Taf13 were shown to interact with TBP and compete with the N-terminus of Taf1 for TBP binding[31]. Our CXMS data support such a dynamic association by revealing that the same TBP residues crosslink with both Taf11-Taf13 and the Taf1 N-terminus. Taf4, a component of the Twin lobe, is crucial for TFIIA binding to TFIID as evidenced by deletion analysis revealing Taf4 sequences next to the HF domain as important for TFIIA-TFIID interaction[47]. A yet to be resolved dynamic interaction network between TFIIA, TBP, subunits of the Twin lobe and the transcription activators Rap1 is functionally important to regulate the expression of ribosomal genes[48]. In humans, the transactivation domain of the oncogenic transcription factor MYB binds directly to the HFDs of Taf4/12 to drive the expression of genes involved in the development of Acute Myeloid Leukemia[49] suggesting an evolutionary conserved function.

## Methods

**Preparative scale production of TFIID.** The TFIID complex was purified from nuclear extracts of a budding yeast *Komagataella phaffii* strain using a streptavidin-binding peptide (SBP) affinity tag placed at the C-terminus of the Taf2 subunit (Supplementary Fig. 1). 2 L of yeast cells were grown at 24 °C with glycerol as carbon source and harvested when OD$_{600\ nm}$ reached 12–15. Cells were washed in water and then treated with 10 mM DTT. The cell wall was digested by addition of lyticase and spheroplasts were pelleted at 5500 g for 20 min. All further steps were performed at 0 to 4 °C. Protease inhibitors were added to all buffers. Spheroplasts were disrupted by suspension in a hypotonic buffer (15–18% Ficoll 400, 0.6 mM MgCl$_2$, 20 mM K-phosphate buffer pH 6.6) using a ULTRA-TURRAX disperser. Sucrose (0.1 M) and MgCl$_2$ (5 mM) were then added. Nuclei (and some debris) were pelleted at 33,000×g for 37 min, resuspended in a wash buffer (0.6 M Sucrose, 8% PVP, 1 mM MgCl$_2$, 20 mM phosphate buffer pH 6.6) and pelleted again at 34,000×g for 50 min. Nuclei were resuspended in extraction buffer (40 mM HEPES pH 8.0, 300 mM potassium acetate, 20% sucrose, 10 mM MgCl$_2$, 2 mM EDTA, 5 mM DTT) with 20 strokes using a tight pestle in a dounce homogenizer. Following 30 min of incubation, debris were precipitated at 33,000×g for 38 min. The supernatant was collected and 1–2% PEG 20,000 added in order to precipitate some remaining organelles and membrane parts by a short centrifugation step at 33,000×g for 10 min. The PEG 20,000 concentration was then increased to 5.8% and TFIID precipitated in a second short centrifugation step. The pellet was solubilized in a minimal volume and avidin was added to block endogenously biotinylated proteins. The suspension was incubated with streptavidin beads for 4 h in buffer A (40 mM HEPES pH 8.0, 250 mM potassium acetate, 10% sucrose, 2 mM MgCl$_2$, 2 mM DTT) washed 5 times and eluted with buffer A containing 10 mM biotin. The eluate was concentrated with Millipore Amicon-Ultra (50KDa cut-off) and spun in a 10–30% sucrose gradient with buffer B (20 mM HEPES pH 8.0, 150 mM Potassium acetate, 2 mM DTT, 3 mM MgCl$_2$, 0.2 mM EDTA) in rotor SW60 (39,600 rpm for 15.5 h.). TFIID was fractionated at approx. 25% sucrose and concentrated with Amicon-Ultra to ∼ 1 mg/ml.

**Cross-linking and mass spectrometry.** 50 µg of purified TFIID was cross-linked by addition of 3 mM BS3 (Thermo-Scientific) for 2 h at 25 °C. Samples were digested with trypsin, and the resulting peptides were fractionated by strong cation exchange (SCX) chromatography and analyzed by MS (Orbitrap Fusion). The crosslinked peptides were identified by searching the MS data against a database composed of *K.p.* TFIID subunit sequences using two different algorithms: pLink and in-house designed Nexus (available upon request) as described before[50]. A 5% of false discovery rate (FDR) was used for both pLink and Nexus searches. The circular crosslinking map was generated using ProXL[51].

**Formation of promoter-bound complexes.** The yeast *S. cerevisiae* TFIIA used for stabilization of TFIID-DNA complexes was recombinantly expressed in *E.coli*, purified from inclusion bodies and reconstituted as described earlier[52].

Promoter DNA fragments were obtained by annealing of equimolar amounts of complementary oligonucleotides at a final concentration of 10 µM in 20 mM Tris-HCl; 2 mM MgCl$_2$, 50 mM KCl by heating the mixture to 95 °C for 5 min and cooling slowly down to room temperature. The 105 nucleotide fragment of pGAP promoter used for EM (5′-gacgcatgtcatgagattattggaaaccaccagaatcgaatataaaaggc gaacacctttcccaattttggtttctcctgacccaaagactttaaatttaattta-3′) contained 20 nucleotides downstream of the TSS and 40 nucleotides upstream of TATA-box. For the gel-shift experiments fluorescently labeled DNA was used: pGAP (5′-[6FAM]tgtcatga gattattggaaaccaccagaatcgaatataaaaggcgaacacctttcccaattttggtttctcctgacccaaagactt-taaatttaattta-3′).

For the gel-shift experiments 0.4 µM TFIID was incubated with 0.2 µM of double-strand DNA fragment in presence or absence of twofold molar excess of TFIIA in the buffer containing 15% sucrose, 150 mM Potassium acetate, 20 mM Hepes pH8.0, 5 mM MgCl$_2$. Protein-DNA complexes were formed for 30 min at room temperature and loaded on a native 1% agarose, 1.5% acrylamide gel containing 5% glycerol and 5 mM MgCl$_2$ in Tris-Glycine buffer. Gels were analysed using Typhoon FLA9500 imager (GE Healthcare Life Sciences).

For the EM-studies 0.4 µM TFIID was incubated with two-fold excess of TFIIA and 2.5–3-fold excess of pGAP promoter DNA to have all TFIID molecules bound to DNA.

**Cryo-EM sample preparation and data acquisition.** Freshly purified TFIID or assembled TFIID-TFIIA-DNA complexes were cross-linked with glutaraldehyde (final concentration 0.1%) for 30 min on ice. After the cross-linking reaction was stopped, samples were dialyzed using VSWP MF-membrane Filters (Millipore) to remove sucrose. 3 µl of sample was applied onto a holey carbon grid (Quantifoil R2/2 and UltrAuFoil R1.2/1.3 300 mesh) rendered hydrophilic by a 30 s glow-discharge in air (2.5 mA current at 1.8 × 10–1 mbar). The grid was blotted for 2.5 s (blot force 5) and flash-frozen in liquid ethane using Vitrobot Mark IV (FEI) at 4 ° C and 95% humidity.

Images were acquired on a Cs-corrected Titan Krios (FEI) microscope operating at 300 kV in nanoprobe mode using the serialEM software for automated data collection[53]. Movie frames were collected in the case of holo-TFIID on a 4k × 4k Falcon 2 direct electron detector at a nominal magnification of 59,000 which

yielded a pixel size of 1.1 Å. Seven movie frames were recorded at a dose of 7 electrons per Å2 per frame corresponding to a total dose of 60 e/Å2. In the case of TFIID-TFIIA-DNA the movies were recorded on a 4k × 4k Gatan K2 summit direct electron detector in super-resolution mode at a nominal magnification of 105,000, which yielded a pixel size of 0.55 Å. Forty movie frames were recorded at a dose of 1.32 electrons per Å2 per frame corresponding to a total dose of 52.8 e/Å2, but only the last 38 frames were kept for further processing.

**Initial reference generation.** Grids containing frozen-hydrated TFIID sample were subjected to tomographic acquisition on a Cs-corrected Titan Krios (FEI) microscope operating at 300 kV in nanoprobe mode using the FEI Tomo software. Images were recorded on a Falcon 2 camera at a nominal magnification of 29,000, which resulted a pixel size of 3.8 Å. Tomographic images were taken with a tilt from −60° to +60° with an increment of 1˚. Tomograms were reconstructed in IMOD and sub-tomograms containing single TFIID particles were extracted using the same software[54]. Maximum-likelihood based sub-tomogram alignment and classification was performed in Xmipp.

**Image processing.** Movie frames were aligned, dose-weighted, binned by 2 and averaged using Motioncor2[55] to correct for beam-induced specimen motion and to account for radiation damage by applying an exposure-dependent filter. Non-weighted movie sums were used for Contrast Transfer Function (CTF) estimation with Gctf[56] program, while dose-weighted sums were used for all subsequent steps of image processing. After manual screening, images with poor CTF, particle aggregation or ice contamination were discarded. About 6,000 TFIID particles were picked manually using the e2boxer program of EMAN2[57] and subjected to 2D classification in Relion[58]. Representative class average images showing TFIID in different orientations were then used as references for auto-picking with Gautomatch (http://www.mrc-lmb.cam.ac.uk/kzhang/Gautomatch/) for both datasets. Several cycles of automatic picking followed by 2D and 3D classification were performed, yielding datasets of 155,620 particles for holo-TFIID. The same procedure was applied to the TFIID-TFIIA-DNA dataset along with random-phase classification (30) resulting in 180,823 particle images. These datasets were analyzed in Relion 1.4 and Relion 2 according to standard protocols. The structures were refined using a low-pass filtered starting model obtained previously by tomography followed by sub-tomogram averaging. Global resolution estimates were determined using the FSC = 0.143 criterion after a gold-standard refinement. Local resolution was estimated with ResMap[59].

Three-dimensional classification of the entire dataset could not clearly separate distinct conformations of TFIID complex. Therefore we carried out a focused refinement of the separate lobes using the masked lobes as references.

**Model building.** Homology models of protein domains with known atomic structures were made using I-TASSER[60] namely the amino-peptidase domain of Taf2; the HEAT repeats of Taf6, histone-fold domains of Taf4, Taf6, Taf9, Taf11, Taf12 and Taf13, the WD repeats and the N-terminal domain of Taf5. Initial rigid body docking of the homology models into the cryo-EM map of TFIID was performed using ADP-EM[61]. A top scoring solution was found to be in close agreement with previous manual docking. The carbon alpha traces of the models were manually corrected in COOT[62] according to density, taking into account the secondary structure prediction as obtained from Phyre2[63]. In a few cases (putative part of Taf2 C-terminus, putative long helices in Taf8) alpha helices were placed in a density that was not occupied by the homology models and the helical domain was attributed to a subunit after considering density continuity, 2D predictions, XL/MS and additional published data.

Model geometry was then idealized using phenix.geometry_minimization with secondary structure restraints[64]. All display images were generated using UCSF Chimera[65] and ChimeraX[66].

## Data availability

The Mass spectrometry raw date were deposited to PRIDE with accession code PXD011092. The cryo-EM maps have been deposited in the 3D-EM database (www.ebi.ac.uk/pdbe) with accession codes EMD-0249, EMD-0250, EMD-0251, EMD-0253, EMD-0254 and EMD-0255. The model coordinates were deposited in the PDB database with accession code 6HQA.

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

## Acknowledgements

We are grateful to I. Berger for providing the X-ray structure of the TAF5-6-9 before publication. This work was supported by the Institut National de la Santé et de la Recherche Médicale (INSERM), the Centre National pour la Recherche Scientifique (CNRS), the Association pour la Recherche sur le Cancer (ARC), the Ligue contre le Cancer, the Fondation pour la Recherche Médicale (FRM), the ANR-15-CE11-0022-01 and the grant ANR-10-LABX-0030-INRT, a French State fund managed by the Agence Nationale de la Recherche under the frame program Investissements d'Avenir ANR-10-IDEX-0002-02. We acknowledge the support and the use of resources of the French Infrastructure for Integrated Structural Biology FRISBI ANR-10-INBS-05 and of Instruct-ERIC. This work was supported by U.S. NIH grants P50 GM076547 RO1 GM110064 to J.R. We thank Laszlo Tora for discussion.

## Author contributions

P.S., G.P. and A.B.S. designed the study; A.B.S originated the purification procedure; A.B.S and O.K. produced, purified and characterized the *K. phaffii* TFIID complex; A.B.S and O.K. treated purified TFIID before deposition on grids. G.P. froze grids, collected and analyzed cryo-EM data, calculated and refined the EM density; J.R. and J.L. carried out and interpreted the mass-spectrometry experiments, all authors contributed in preparing figures and writing the manuscript.

## Additional information

**Competing interests:** The authors declare no competing interests.

