## [Peer Review File · Nature Communications]

REVIEWERS' COMMENTS:

Reviewer #1 (Remarks to the Author):

The manuscript of Papai and Schultz describes a timely and interesting study towards the elucidation of the structure of DNA-bound TFIID. This work is worthy of publication for the arguments I provided before.

I have only one reservation before a full recommendation to publish the current manuscript. This (still) concerns their models of dynamic DNA-binding by TFIID, for which very little evidence is presented in this study. In fact, the EM analysis showed that 'TFIID complex was not drastically reorganized upon promoter DNA' (page 4). But the authors base changes in TFIID conformation upon DNA binding on the electrophoretic mobility analyses of Fig. S2 and S3, which do not look very convincing and remains the weakest part of the manuscript.

The authors observe an additional band (marked by an asterisk) upon addition of TFIIA, and label this as the TFIIA-TBP-DNA complex. It has been well-documented that TFIIA-stimulated DNA-binding by TBP is TATA-dependent. However, in this study the formation of this faster-migrating complex is not TATA-dependent (compare lane 8 with lane 4 in Fig. S2b), and forms on DNA fragments lacking any TATA sequences (compare lane 8 with lane 4 in Fig. S3b). The statement in the text that 'These experiments show that TFIID binds to non-promoter DNA and that this interaction is stimulated when a TATA-box is present' is not clear from the presented data, which indicates only small differences in the band labeled by the asterisk.

Clearly, this complex requires a more extensive characterization, including DNA protection or methylation interference assays, DNA-protein cross-linking, super-shift, parallel reactions with recombinant TBP from *K. phaffii*, etc. As an alternative to this extensive characterization, the authors may choose to remove their speculation on TBP-dissociation from TFIID, sliding of TFIID along promoter DNA, etc. from their discussion. Please note that this minor aspect of the study now takes a full page (~40%) of the discussion section on pages 7 and 8.

Minor points:

- lettering in Fig. 1B falls out of the boxes: P-rich in Taf6, α -helix in Taf11, HFD in Taf10, etc.
- the TAF7 interaction domain in Fig. S1 is (still) mis-labeled as 'HAT'.
- page 6: 'TAF6-9 and TAF4-12' should be 'TAF6-9 and TAF4-12'.
- page 8: 'conserved core promoter motive' should be 'conserved core promoter motif'.

Reviewer #3 (Remarks to the Author):

The revised manuscript adequately addresses the comments of the reviewers. In addition, the figures have been improved. It should be fine for publication in Nature Communications.

Comments

Page 3 bottom paragraph, page 8 middle paragraph and Supplementary figures 2 and 3. The authors conclude that the faster migrating band contains TBP and TFIIA based on mass spec analysis of a gel slice. It is possible, however, that a substantial fraction of the lane contains TBP and TFIIA. The mass spec assay does not show that TBP and TFIIA are specifically enriched at the location of the faster migrating band. One possible control would be to perform the gel shift and mass spec analysis with a sample run in parallel without the DNA probe. This would reveal whether or not there are background

levels of TBP and TFIIA in the gel.

It should also be noted that the experiments involve the use of extremely high concentrations (400 nM) of TFIIID. The high concentrations of TFIIID, TFIIA, and DNA might explain the lack of sequence specificity of the binding.

Typos

Page 4 middle paragraph - "supported by multiple crosslinks", "at the base of this yeast-specific" and "close to the stalk base".

Page 8 middle paragraph - "TFIIA results in a faster migrating TFIIID-DNA species" and "In metazoans, Taf1", "core promoter motifs, and were shown to form a downstream promoter binding module that interacts with the".

Supp. Fig. 9 legend - "c, Modelled ppTFIIID subunits"

Response to reviewers' comments

We wish to thank the reviewers for their constructive comments to our manuscript. In our response below, we address point-by-point all comments raised by the Referees. We have revised our manuscript as requested.

Our responses to each referees' comments, point by point, are detailed in the insertions in the text below in blue.

Reviewer #1 (Remarks to the Author):

The manuscript of Papai and Schultz describes a timely and interesting study towards the elucidation of the structure of DNA-bound TFIID. This work is worthy of publication for the arguments I provided before.

I have only one reservation before a full recommendation to publish the current manuscript. This (still) concerns their models of dynamic DNA-binding by TFIID, for which very little evidence is presented in this study. In fact, the EM analysis showed that 'TFIID complex was not drastically reorganized upon promoter DNA' (page 4). But the authors base changes in TFIID conformation upon DNA binding on the electrophoretic mobility analyses of Fig. S2 and S3, which do not look very convincing and remains the weakest part of the manuscript. The authors observe an additional band (marked by an asterisk) upon addition of TFIIA, and label this as the TFIIA-TBP-DNA complex. It has been well-documented that TFIIA-stimulated DNA-binding by TBP is TATA-dependent. However, in this study the formation of this faster-migrating complex is not TATA-dependent (compare lane 8 with lane 4 in Fig. S2b), and forms on DNA fragments lacking any TATA sequences (compare lane 8 with lane 4 in Fig. S3b). The statement in the text that 'These experiments show that TFIID binds to non-promoter DNA and that this interaction is stimulated when a TATA-box is present' is not clear from the presented data, which indicates only small differences in the band labeled by the asterisk.

Clearly, this complex requires a more extensive characterization, including DNA protection or methylation interference assays, DNA-protein cross-linking, super-shift, parallel reactions with recombinant TBP from *K. phaffii*, etc. As an alternative to this extensive characterization, the authors may choose to remove their speculation on TBP-dissociation from TFIID, sliding of TFIID along promoter DNA, etc. from their discussion. Please note that this minor aspect of the study now takes a full page (~40%) of the discussion section on pages 7 and 8.

We agree with the referee that the conclusion for DNA sliding and TBP deposition stems from indirect evidence and needs to be sustained by more direct experiments. We therefore removed from the results, as well as from the discussion, the speculative conclusions about TFIID-DNA interactions.

Minor points:

- lettering in Fig. 1B falls out of the boxes: P-rich in Taf6, α -helix in Taf11, HFD in Taf10, etc.
- the TAF7 interaction domain in Fig. S1 is (still) mis-labeled as 'HAT'.
- page 6: 'TAF6-9 and TAF4-12' should be 'TAF6-9 and TAF4-12'.

-page 8: 'conserved core promoter motive' should be 'conserved core promoter motif'.

We corrected the typos in the text and changed the figure labelling accordingly.

Reviewer #3 (Remarks to the Author):

The revised manuscript adequately addresses the comments of the reviewers. In addition, the figures have been improved. It should be fine for publication in Nature Communications.

Comments

Page 3 bottom paragraph, page 8 middle paragraph and Supplementary figures 2 and 3. The authors conclude that the faster migrating band contains TBP and TFIIA based on mass spec analysis of a gel slice. It is possible, however, that a substantial fraction of the lane contains TBP and TFIIA. The mass spec assay does not show that TBP and TFIIA are specifically enriched at the location of the faster migrating band. One possible control would be to perform the gel shift and mass spec analysis with a sample run in parallel without the DNA probe. This would reveal whether or not there are background levels of TBP and TFIIA in the gel.

It should also be noted that the experiments involve the use of extremely high concentrations (400 nM) of TFIID. The high concentrations of TFIID, TFIIA, and DNA might explain the lack of sequence specificity of the binding.

We thank the reviewer for careful evaluation of our revised manuscript. We removed from the manuscript the speculations about the interaction between TFIID, TFIIA and DNA as it requires additional data to be firmly established.

Typos

Page 4 middle paragraph - "supported by multiple crosslinks", "at the base of this yeast-specific" and "close to the stalk base".

Page 8 middle paragraph - "TFIIA results in a faster migrating TFIID-DNA species" and "In metazoans, Taf1", "core promoter motifs, and were shown to form a downstream promoter binding module that interacts with the".

Supp. Fig. 9 legend - "c, Modelled ppTFIID subunits"

We corrected the typos in the text and in Suppl. Fig. 9 legend.